# Influence of Adherence to the Mediterranean Diet and Level of Physical Activity with Liver Steatosis in People Aged > 50 Years and with a BMI > 25 kg/m^2^: Association with Biochemical Markers

**DOI:** 10.3390/nu16131996

**Published:** 2024-06-23

**Authors:** Laura Cano-Lallave, Jaime Ruiz-Tovar, Lucia Martin-de-Bernardo, Malena Martinez-Oribe, Cristina Rodriguez-Obispo, Sara Carrascosa-Corrochano, Ana Martín-Nieto, Isabel Baeza, Marta Gonzalez-Ramos, Marta Benito, Isabel Olazabal

**Affiliations:** 1Department of Health Sciences, School of Medicine, Alfonso X University, 28691 Madrid, Spain; lcanolal@uax.es; 2San Juan de Dios Foundation, 28015 Madrid, Spain; amartinn@comillas.edu (A.M.-N.); ibaeza@comillas.edu (I.B.); mogonzalezr@comillas.edu (M.G.-R.); mbenitom@comillas.edu (M.B.); 3Health Sciences Department, San Juan de Dios School of Nursing and Physical Therapy, Comillas Pontifical University, 28015 Madrid, Spain; 4Department of Biomedicine, Alfonso X University, 28691 Madrid, Spain; lmartind@myuax.com (L.M.-d.-B.); morivmar@myuax.com (M.M.-O.); cobisrod@myuax.com (C.R.-O.); scarrcor@myuax.com (S.C.-C.); iolazola@uax.es (I.O.)

**Keywords:** hepatic steatosis, non-alcoholic fatty liver disease, NAFLD, cardiovascular risk factors, biochemical markers, Mediterranean diet, adherence, physical activity, obesity

## Abstract

Background: The main objective of this study is to determine the accuracy of different biochemical markers of hepatic steatosis and to correlate liver steatosis with adherence to the Mediterranean diet and level of physical activity. Methods: A cross-sectional study was carried out, including subjects over 50 years of age, with a BMI > 25 kg/m^2^, but excluding any patient with documented hepatic pathology other than hepatic steatosis. Participants were divided into two groups: patients with hepatic steatosis diagnosed by ultrasound (SG) and a control group of individuals without hepatic steatosis (CG). The level of physical activity was recorded by the IPAQ-SF questionnaire and the adherence to the Mediterranean diet was recorded using the PREDIMED questionnaire. Biochemical markers analyzed included the Hepatic steatosis index (HSI), AST-to-Platelet ratio (APRI) and Fibrosis-4 (FIB-4). Results: A total of 116 patients were included, 71 belonging to the SG and 45 to the CG. A total of 58.6% of the patients showed low adherence to the Mediterranean diet, 35.4% moderate adherence and 6% high adherence. The median estimated physical activity was 495 METS, with most participants reporting light activity. In the SG, significantly higher HSI values were observed (*p* < 0.001). A cut-off point of a HSI of 40 was established, with a sensitivity of 73.2% and a specificity of 65.8%. Significantly higher FIB-4 values (*p* = 0.039) were also observed in the SG. A cut-off point of FIB-4 was set at 0.27, with a sensitivity of 69% and a specificity of 57.9%. Patients in the SG showed lower scores in the PREDIMED. Patients in the SG tended to show lower METS scores. However, the higher number of patients with intense activity in the CG group stands out (*p* = 0.008). Conclusions: The HSI and FIB-4 showed a significant correlation with liver steatosis. Hepatic steatosis is associated with low adherence to the Mediterranean diet and patients with hepatic steatosis tended to have lower METS scores.

## 1. Introduction

Non-alcoholic fatty liver disease (NAFLD) is a clinical-histopathological entity that occurs in patients who do not consume harmful amounts of alcohol. This entity encompasses a broad spectrum of histopathological liver changes such as hepatic steatosis, steatohepatitis, fibrosis and cirrhosis [1].

Hepatic steatosis is an accumulation of triglycerides within hepatocytes when the mechanisms for their synthesis exceed the mechanisms for their disposal. Steatosis can develop into steatohepatitis when liver cell death and severe inflammation occur. The latter is also known as Non-Alcoholic Steatohepatitis (NASH), which can progress to cirrhosis through the progressive accumulation of fibrous scar tissue. The development of cirrhosis in NAFLD implies an increase in the annual incidence of primary liver cancer by up to 3% [1].

The main cause of hepatic steatosis in developed countries is obesity and being overweight, with poor dietary habits and a sedentary lifestyle being the main conditioning factors. Currently, the increase in time spent in sedentary leisure activities and dietary habits focused on processed foods has led to an increase in the incidence and prevalence of overweight and obesity from an early age in our society [1,2].

Obesity plays an important role in hepatic steatosis as it stimulates fat accumulation in hepatocytes by altering intestinal microbiota and intestinal permeability. This altered intestinal permeability leads to reduced intestinal function by increasing the liver’s exposure to products from the gut, which stimulate liver cells to produce inflammatory mediators that inhibit the action of insulin. This situation of insulin resistance leads to hyperglycemia. The net result is the accumulation of triglycerides in hepatocytes, i.e., steatosis [2,3,4].

Diet and exercise also play an important role in the development of hepatobiliary disease as lifestyle, age and body mass index (BMI) are risk factors that modify the prevalence of NAFLD [3]. In fact, lifestyle and dietary changes are the basis for the treatment of NAFLD. The loss of at least 3–5% of body weight improves hepatic steatosis, in the same way as a low-carbohydrate diet [1].

Early detection of the severity of liver damage in NAFLD is key to providing patients with a prognosis as well as therapeutic recommendations. Moreover, this early detection can improve the outcome of their liver disease and allows patients with steatosis without inflammation or hepatocyte cell death (simple steatosis) to be distinguished from those with NASH and NASH patients who have advanced fibrosis to be identified. There are different strategies for stratification, which can be divided into invasive tests such as liver biopsy, the gold standard test for both diagnosis and stratification of NAFLD, and non-invasive tests such as imaging techniques (ultrasound, computed tomography (CT) and magnetic resonance imaging (MRI)), although the latter cannot determine which individuals with NASH have NAFLD. The increasing prevalence of NAFLD makes it necessary to find a less invasive technique than liver biopsy [1,5].

Currently, some biochemical markers [6,7] have been proposed to evaluate hepatic steatosis, the risk of progression to NASH and the degree of liver fibrosis. These tests have proven to be very useful in the early detection of hepatic steatosis. They constitute useful tools for widespread use as they are available to everyone and, as stated by the European Association for the Study of the Liver (EASL) [5], they can be used to test for liver fibrosis and steatosis in primary care and secondary prevention. Furthermore, they are also considered to be highly reproducible and applicable. These include the Hepatic Steatosis Index (HSI), the Liver Fat Score (LFS), the Fibrosis-4 (FIB-4), the Non-Alcoholic Fatty Liver Disease Fibrosis Score (NAFLD-FS) or the AST/Platelet Ratio Index (APRI), which are based on biochemical parameters obtained in routine blood analyses (AST, ALT, platelet count or albumin) and demographic and anthropometric parameters (age, sex or BMI) to estimate values that indicate the likelihood of NASH or liver fibrosis [4,6,8,9].

The aim of this study is to determine the association between the presence of hepatic steatosis and the adherence to a Mediterranean diet pattern, as well as to evaluate the correlation of hepatic steatosis with the level of physical activity in a population of overweight or obese people aged over 50 years. As previously mentioned, excess weight is the main cause of hepatic steatosis in our environment and its prevalence increases from the fifth decade of life onwards, both in men and women. For this reason, we chose to study a population in this age range and with this excess weight [1,2]. Likewise, the association between biochemical markers of hepatic steatosis and ultrasound diagnosis of liver steatosis was also evaluated, determining the diagnostic accuracy of these biochemical indicators in the studied population.

## 2. Materials and Methods

A cross-sectional and observational study was carried out from April to June 2023 in the Primary Care centers La Estación and Puente del Arzobispo (Toledo, Spain). The study population included patients aged over 50 years, with a BMI > 25 kg/m^2^, with an ultrasound diagnosis in the previous 6 months assessing the presence of hepatic steatosis, and with a blood test in the 3 months prior to the inclusion in the study, including the assessment of biochemical data necessary for the calculation of biochemical markers of hepatic steatosis.

Exclusion criteria were the following: the presence of documented liver pathology other than hepatic steatosis, alcohol abuse, previous infection with hepatitis C virus, symptomatic biliary pathology, intellectual disability or inability to understand the questionnaires, physical disability preventing age-appropriate physical activity and psychiatric eating disorders.

Non-probability convenience sampling was used. The sample size was calculated using the G-Power program, which estimates sample size based on a 95% confidence level, a 5% margin of error in estimation and an additional 10% overestimation to compensate for lack of interest in participation [10]. A minimum sample size of 40 participants was calculated for both groups to obtain clinically relevant results, with no upper limit set for either the group of patients with hepatic steatosis or the control group without steatosis.

Participants were divided into 2 groups:Patients with ultrasound-diagnosed hepatic steatosis: Patients aged over 50 years with a BMI > 25 kg/m^2^, with an ultrasound diagnosis of hepatic steatosis and who attended primary care.Control group of individuals without hepatic steatosis: To assess the effect of adherence to the Mediterranean diet and physical activity, a group of patients aged over 50 years with a BMI > 25 kg/m^2^ but without ultrasound evidence of hepatic steatosis and who attended the same primary care centers were also included in the study.

### 2.1. Questionnaires Used

Adherence to the Mediterranean diet was analyzed using the PREDIMED [11] questionnaire, which has been validated within the Spanish population. It includes a series of questions about dietary habits and food consumption, with each answer receiving a score. Depending on the score obtained, patients were classified as having low, medium or high adherence to the Mediterranean diet.

Weekly physical activity was assessed using the IPAQ-SF [12] questionnaire, validated for the Spanish population. Based on their answers, the physical activity performed will be recorded in Metabolic Equivalent of Task (METS) by multiplying each of the values associated with walking, moderate physical activity and vigorous physical activity by the time in minutes and frequency in which they were performed. Patients with a METS score of less than 80 were classified as sedentary, those with a score between 81 and 600 as light activity, those with a score between 601 and 1500 as moderate activity and those with a score greater than 1501 as vigorous activity.

### 2.2. Calculation of Biochemical Markers of Hepatic Steatosis

The biomarkers used to predict the presence of hepatic steatosis and fibrosis from the patients’ analytical data were as follows:Hepatic Steatosis Index (HSI) = 8 × ALT/AST + BMI (+2 if the patient has DM2; +2 in women).AST to platelet ratio index (APRI) = (AST/ALT × 100)/platelet count (10^9^/L).Fibrosis-4 (FIB-4) = (age [years] × AST)/(platelet count × ALT)

### 2.3. Variables

The variables studied included demographic variables (age and sex), comorbidities (hypertension, type 2 diabetes mellitus, dyslipidemia and hepatic steatosis determined by ultrasound), anthropometric variables and body composition (weight, BMI and adiposity). Adiposity was assessed using the CUN-BAE formula (Clinica Universitaria de Navarra-Body Adiposity Estimation), validated for the Spanish population, which allows the percentage of body fat to be estimated from the age, sex, weight and height of each individual. This value is calculated using the following operation:Adiposity (%) = −44.988 + (0.503 × age) + (10.689 × sex) + (3.1723 × IMC) − (0.026 × IMC^2^) + (0.181 × IMC × sex) − (0.023 × IMC × age) − (0.005 × IMC^2^ × sex) + (0.00021 × IMC^2^ × age)
Sex: man = 0 and woman = 1; Age in years.

In addition, adherence to the Mediterranean diet was assessed quantified using the PREDIMED questionnaire, and physical activity was assessed using the IPAQ-SF scoring protocol.

Finally, the biochemical markers HIS, APRI and FIB-4 were calculated.

### 2.4. Statistical Methodology

Quantitative variables that followed a normal distribution were defined by mean and standard deviation. For variables that did not follow a Gaussian distribution, the median was used instead of the mean as a measure of centralization. Discrete variables were defined by percentage and number of cases.

The following methods were used to analyze the variables:Comparison between qualitative variables: When comparing two discrete variables, the Chi-square test was used. When the expected value in any of the boxes of the contingency table was less than 5, it was necessary to use Fisher’s exact test. Relative risk was used to estimate the magnitude of the association.Comparison of two normal quantitative variables: Pearson’s comparison method was used. If one or both variables did not have a Gaussian distribution, Spearman’s test was used.Comparison of two independent means: Student’s *t*-test (Mann–Whitney U test for non-Gaussian variables).Correlation between quantitative variables: Pearson correlation coefficient (Spearman correlation coefficient for non-Gaussian variables).

To establish cut-off points as diagnostic methods for quantitative variables, ROC curve analysis was performed to determine the area under the curve as well as the sensitivity and specificity of the established cut-off point.

Values *p* < 0.05 were considered statistically significant. Data processing and analysis were performed using SPSS 28.0 statistical software for Windows.

### 2.5. Ethical Aspects

This study was evaluated and approved by the Research Ethics Committee of Alfonso X University. The database will be anonymized in accordance with the criteria of the Organic Law 3/2018 on Data Protection and in compliance with the provisions of Regulation (EU) 2016/679, preventing the direct and indirect identification of participants and the processing of their data. Likewise, during the project, the guidelines or regulations of the Declaration of Helsinki have been followed, especially those approved at the 64th General Assembly in Fortaleza (Brazil), as well as the national legislation in force regarding the analysis of persons and their protection and confidentiality.

## 3. Results

A total of 116 patients were included; 71 belonged to the group with hepatic steatosis (steatosis group-SG) and 45 belonged to the group without ultrasound evidence of steatosis (control group-CG). A total of 65 patients in the total sample were men (56%) and 51 women (44%). The prevalence of hepatic steatosis was 2.2 times higher in men than in women (RR 2.2; 95% CI (1.1–4.6); *p* = 0.045). The mean age of the sample was 68.1 ± 12.2 years. The mean age of patients in the SG group was significantly older than the one of the patients in the CG group (difference 8.7 years; 95% CI (4.3–13.1); *p* = 0.000).

The most common comorbidity was hypertension (50%), followed by dyslipidemia (40.5%) and type 2 diabetes mellitus (33.1%). A previous diagnosis of hypertension implies a 2.6-fold increased risk of hepatic steatosis (95% CI 1.2–5.7; *p* = 0.022), a diagnosis of dyslipidemia a 4.6-fold increased risk of hepatic steatosis (95% CI 1.9–10.9; *p* = 0.000) and diabetes mellitus a 9.7-fold increased risk of hepatic steatosis (95% CI 3.4–27.8, *p* = 0.000).

The study found that 19.1% of the patients were active smokers. In the hepatic steatosis group, 28.2% of patients were smokers compared to only 4.5% in the control group. Therefore, smoking increased the risk of developing hepatic steatosis by a factor of 8.3 (95% CI 1.8–37; *p* = 0.001).

### 3.1. Anthropometric Variables

Weight, BMI and adiposity were significantly higher in the SG group (Table 1).

The obesity rate was significantly, higher in the SG than in the CG (Table 2).

### 3.2. Adherence to the Mediterranean Diet

The median score on the PREDIMED test was 7 points (range 5–13). On this basis, patients were classified as having low, moderate or high adherence to the Mediterranean diet. It is to be noted that 58.6% of patients had low adherence to the Mediterranean diet, 35.4% had moderate adherence and only 6% had high adherence (Figure 1).

### 3.3. Physical Activity Level

The median estimated physical activity was 495 METS (range 14–7200). Categorizing the METS into sedentary and light, moderate and vigorous activity, it was observed that most participants reported light physical activity (Table 3).

### 3.4. Biochemical Markers of Hepatic Steatosis

The mean HSI was 41.8 ± 5.9. Significantly higher HSI values were observed in the group with hepatic steatosis (mean difference 5.35; 95% CI (3.2–7.5); *p* < 0.001). Using the ultrasound diagnosis of steatosis as the reference test, a cut-off value of 40 for the HSI was established with a sensitivity of 73.2% and a specificity of 65.8% (AUC 0.741; 95% CI (0.643–0.839); *p* = 0.000) (Figure 2). The cut-off value was established according to the ROC curve coordinates to obtain the most accurate values of sensibility and specificity and be the most clinically meaningful.

The mean APRI was 0.52 ± 0.29. There were no statistically significant differences in APRI between the two groups.

The mean FIB-4 value was 0.37 ± 0.24. Significantly higher FIB-4 values were observed in the hepatic steatosis group (mean difference 0.1; 95% CI (0.05–0.2); *p* = 0.039). A cut-off value for FIB-4 of 0.27 gave a sensitivity of 69% and a specificity of 57.9% (AUC 0.655; 95% CI (0.542–0.767); *p* = 0.008). (Figure 3). The cut-off value was established according to the ROC curve coordinates to obtain the most accurate values of sensibility and specificity and be the most clinically meaningful.

The distribution of values for biochemical markers of steatosis in both groups is summarized in Table 4.

### 3.5. Correlation between Hepatic Steatosis and Adherence to the Mediterranean Diet

Patients in the group with hepatic steatosis had a mean PREDIMED test score of 6.6 ± 1.2 compared to 9.3 ± 2.2 in the group without hepatic steatosis (*p* < 0.001).

When the patients were grouped according to their test scores into low, moderate and high adherence to the Mediterranean diet, 80.3% of patients with hepatic steatosis showed low adherence to the Mediterranean diet, while 60% of the control group showed moderate adherence (*p* < 0.001). It is to be noted that in the group with hepatic steatosis, there was no one with high adherence to the Mediterranean diet. These data show that the association between low adherence to the Mediterranean diet and the presence of hepatic steatosis is statistically significant (Figure 4).

The PREDIMED test score only showed a significant inverse correlation with the HSI scores (Pearson −0.190; *p* = 0.048). No significant correlations were found with APRI or FIB-4. Using the HSI cut-off point of 40 as a diagnosis of hepatic steatosis, no statistically significant correlations were found with adherence to the Mediterranean diet.

### 3.6. Association between Hepatic Steatosis and Level of Physical Activity

Patients in the hepatic steatosis group showed a trend toward lower METS scores 728.8 ± 546.3 compared to 1050.3 ± 1442.1 in the group without hepatic steatosis (*p* = 0.092). However, when METS was categorized into light, moderate, vigorous or sedentary physical activity based on the presence of steatosis, the higher number of patients with vigorous activity in the CG (*p* = 0.008) stood out (Figure 5).

The METS score only showed a significant inverse correlation with FIB-4 scores (Pearson −0.279; *p* = 0.003). No significant correlations were found with the APRI or HSI.

Using the established cut-off point of 0.27 for FIB-4 to diagnose hepatic steatosis, no statistically significant correlations were found with the degree of physical activity.

## 4. Discussion

The aim of this study was to evaluate the lifestyle associations of physical activity and dietary habits with NAFLD and other comorbidities. This study has shown that both adherence to the Mediterranean diet and a high level of physical activity are significantly associated with lower severity of hepatic steatosis in overweight or obese people aged over 50.

Our results are consistent with previous studies reporting the benefits of the Mediterranean diet in reducing liver fat and improving metabolic health. Misciagna et al. [13] found that greater adherence to the Mediterranean diet was associated with a lower incidence of NAFLD, which is consistent with our findings. In addition, studies such as Estruch et al. [14] have shown that the Mediterranean diet has beneficial effects on reducing inflammation and improving endothelial function, key factors in the pathogenesis of NAFLD [15].

It is well known that good adherence to the Mediterranean diet is associated with better physical and mental health, greater longevity and less development of obesity or being overweight [15,16,17]. In a published study evaluating different types of diets, the Mediterranean diet was ranked as the number one model diet for reducing cardiovascular risk [18]. The traditional Mediterranean diet is characterized by a high intake of fruits, vegetables, legumes, nuts and cereals, a moderate intake of fish and a low intake of meat and sweets, with olive oil as the main source of lipids of the diet [19]. Many of these foods contain a variety of phytonutrients, including polyphenols and vitamins. The traditional Mediterranean diet is rich in antioxidants such as vitamin E, β-carotene, vitamin C and flavonoids, as well as minerals such as selenium and natural folate [20]. In addition, the Mediterranean diet is considered to be environmentally sustainable [21].

Unfortunately, even in Mediterranean countries, adherence to the Mediterranean diet is decreasing. Modern society has brought about a series of sociological and cultural changes that inevitably affect eating habits and preferences. The amount of time spent shopping and preparing meals has significantly decreased, and instead, processed foods are preferred, which generally involve excessive consumption of foods coming from animals, especially meat and its derivatives, and refined sugars, leading to an increase in saturated fats and cholesterol in the diet [22]. This, combined with an increasingly sedentary lifestyle, has led to a rise in the prevalence of diabetes, being overweight and even morbid obesity [23].

The accumulation of lipids in hepatocytes triggers an inflammatory response that, if sustained over time, leads to NASH. The Mediterranean diet contains anti-inflammatory and antioxidant components that reduce local inflammation. It has been shown that the Mediterranean diet reduces hepatic steatosis and improves insulin sensitivity in an insulin-resistant NAFLD population compared to current dietary recommendations, even without weight loss [24]. Several components of the Mediterranean diet have shown benefits in controlling the pathophysiological mechanisms leading to NAFLD [25]. Some of the most important components of the Mediterranean diet with demonstrated metabolic effects are whole grains with a low glycemic index, foods rich in unsaturated fatty acids, and their content of phytochemical compounds. Whole grains, which are rich in fiber, interact with carbohydrate metabolism and the associated insulin response. Several diabetes clinical trials have demonstrated the beneficial effects of oleuropein or olive leaf extracts on type 2 diabetes mellitus. These studies reported a significant reduction in blood glucose and glycosylated hemoglobin levels. Moreover, oleuropein can promote glucose-stimulated insulin secretion in pancreatic β-cells. Dietary fiber regulates cholesterol and glucose absorption and it also contributes to satiety in order to facilitate weight control. Unsaturated fatty acids improve lipid metabolism at the hepatic level, while phytochemical compounds such as polyphenols have shown moderated anti-inflammatory activity. Inadequate dietary intake of antioxidants can increase the risk of atherosclerotic plaque formation due to alterations in lipoprotein oxidation [26,27,28]. It has been proved that lycopene (carotenoids) from fruits, vegetables, legumes and whole grains have antioxidant activity in hepatocytes, reduce lipid peroxidation and inflammation and improve insulin sensitivity [29].

In a study of the effect of the Mediterranean diet on histologic indicators and imaging tests in NAFLD, it was found that the Mediterranean diet was associated with a reduction in the percentage of intrahepatic lipids and to a lower degree of liver damage, determined by biopsy. Studies evaluating the effect of the diet on hepatic steatosis using MRI showed a reduction in hepatic fat content ranging from 4% to 10% in subjects following the Mediterranean diet. Thus, the Mediterranean diet showed beneficial effects on histological and imaging features of the liver, with an inverse association between adherence to the Mediterranean diet and the severity of liver damage [30].

As the Mediterranean diet does not fit everybody’s preferences, the Atlantic Diet could be a valuable alternative. The fundamental difference between the Mediterranean diet and the Atlantic diet is that the percentage of carbohydrates consumed in the Mediterranean diet decreases significantly and the consumption of high-quality protein increases with the consumption of fish and seafood. These two fresh products are more beneficial than meat in increasing protein intake, as they also have very beneficial unsaturated fats and their caloric intake is lower. To date, there have been few studies specifically looking at the Atlantic diet, so further research is needed to confirm the equivalent benefit of the Mediterranean diet [31].

Regarding physical activity, previous research has shown that regular exercise, particularly aerobic exercise, improves insulin sensitivity, reduces liver fat accumulation, and decreases systemic inflammation. Hallsworth et al. [32] reported that a 12-week aerobic exercise program resulted in a significant reduction in liver fat in patients with NAFLD, supporting the benefits of exercise observed in our study. Therefore, physical activity improves insulin sensitivity and promotes fatty acid oxidation, thereby reducing the amount of fat available to be stored in the liver. Exercise also increases mitochondrial protein expression and improves mitochondrial function, which may help reduce lipid accumulation and inflammation in the liver. Additionally, regular physical activity has been associated with reduced visceral adiposity, which in turn reduces the release of free fatty acids and inflammatory cytokines towards the liver [33].

The clinical implications of these findings are relevant. Promotion of the Mediterranean diet and regular physical activity could be an effective and accessible strategy to reduce the prevalence and severity of NAFLD in at-risk populations, such as overweight or obese individuals aged over 50. Healthcare professionals should consider incorporating dietary and physical activity recommendations into preventive and therapeutic interventions for NAFLD and its associated comorbidities. Implementing interventions to maintain these changes over time is a major challenge [34].

The incidence of NAFLD is increasing worldwide at the same rate as the obesity epidemic [35]. Although liver biopsy remains the gold standard for diagnosis and stratification of NAFLD, it is a resource that cannot be used routinely, so the use of noninvasive tests in clinical practice appears to be much more accessible [36]. Nevertheless, imaging techniques, including ultrasound, consume a large amount of healthcare resources and it would be difficult to plan NAFLD screening strategies using them. Therefore, it is desirable to obtain biochemical markers that, through determinations that can be obtained from routine blood tests, can provide an initial approach to the suspicion of NAFLD. The currently known biomarkers are not specific enough by themselves for the assessment, so a scoring system or predictive model is required. There are a variety of scales to classify hepatic steatosis based on these biomarkers, including HSI, APRI and FIB-4, which are the three parameters analyzed in this study. Biomarkers provide information but are not conclusive and all require critical interpretation of the results and also confirmation by imaging techniques or even histologic confirmation in some cases [6]. In this study, a correlation between HSI and FIB-4 levels and the degree of hepatic steatosis was observed. However, the sensitivity of the biomarkers does allow for screening, but their specificity is too low to replace either imaging or biopsy. Anyway, these associations should be interpreted with caution as this is a cross-sectional study with a relatively small sample size. Future studies are needed to confirm these associations. On the other hand, no significant differences in APRI values were found between the groups, but this may be attributed to the small sample size. Studies with larger numbers of individuals will be necessary to determine the potential value of the APRI biomarker as a screening for hepatic steatosis.

### Limitations

Among the limitations of this study, it should be mentioned that both adherence to the Mediterranean diet and the degree of physical activity were estimated by means of questionnaires. This may lead the patient to overestimate both circumstances. In the present study, it is striking that only two patients described physical activity within the sedentary range, which seems very unlikely in a study population over 50 years of age and with a BMI above 25 kg/m^2^. Therefore, extrapolation of the results obtained should be carried out with caution.

## 5. Conclusions

Ultrasound-detected hepatic steatosis is associated with a poor adherence to the Mediterranean diet. The score obtained in the PREDIMED test showed a significant inverse correlation with the HSI values. Patients with hepatic steatosis were more likely to have a lower METS score as a method of quantifying physical activity, although there was a significantly higher number of patients with vigorous activity in the CG group. The METS score showed a significant inverse correlation with FIB-4 levels.

The HSI was correlated with ultrasound-detected hepatic steatosis. HSI cut-off values above 40 are diagnostic of hepatic steatosis with a sensitivity of 73.2% and a specificity of 65.8%. FIB-4 has also been correlated with ultrasound evidence of hepatic steatosis. FIB-4 cut-off values above 0.27 are diagnostic of hepatic steatosis with a sensitivity of 69% and a specificity of 57.9%. These markers could be used as a screening method for hepatic steatosis in patients older than 50 years with a BMI > 25 kg/m^2^.

## Figures and Tables

**Figure 1 nutrients-16-01996-f001:**
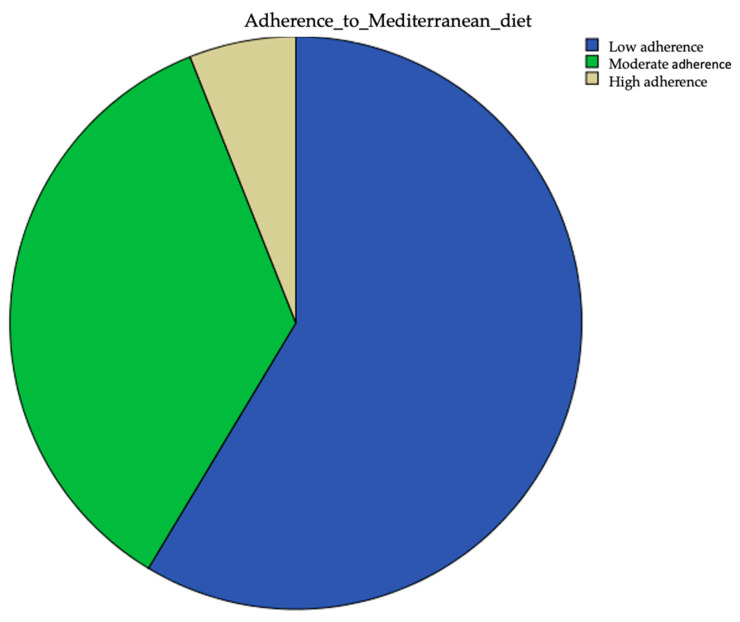
Extent of adherence to the Mediterranean diet.

**Figure 2 nutrients-16-01996-f002:**
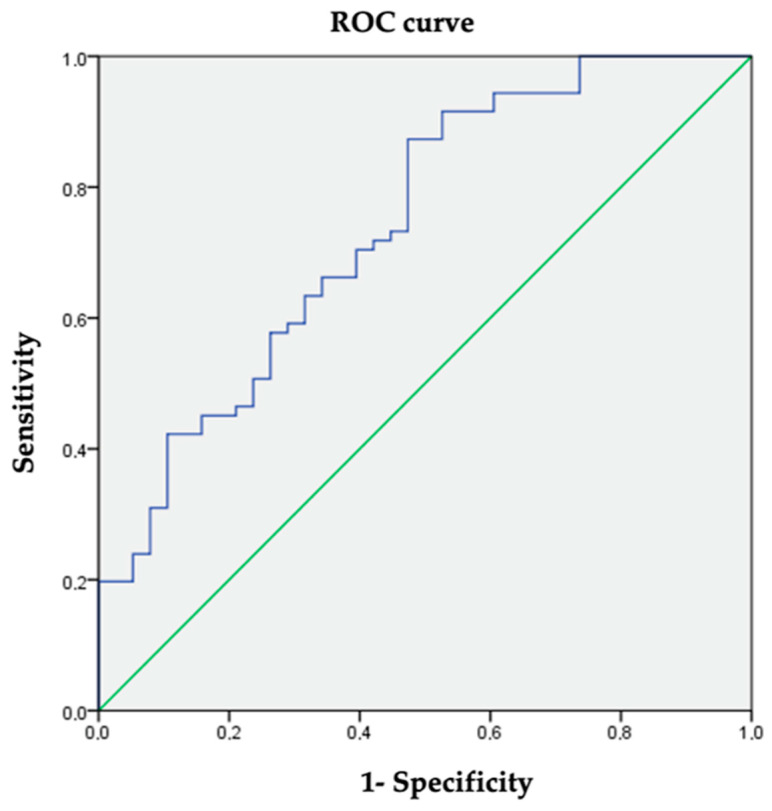
ROC curve coordinates for HSI to diagnose hepatic steatosis.

**Figure 3 nutrients-16-01996-f003:**
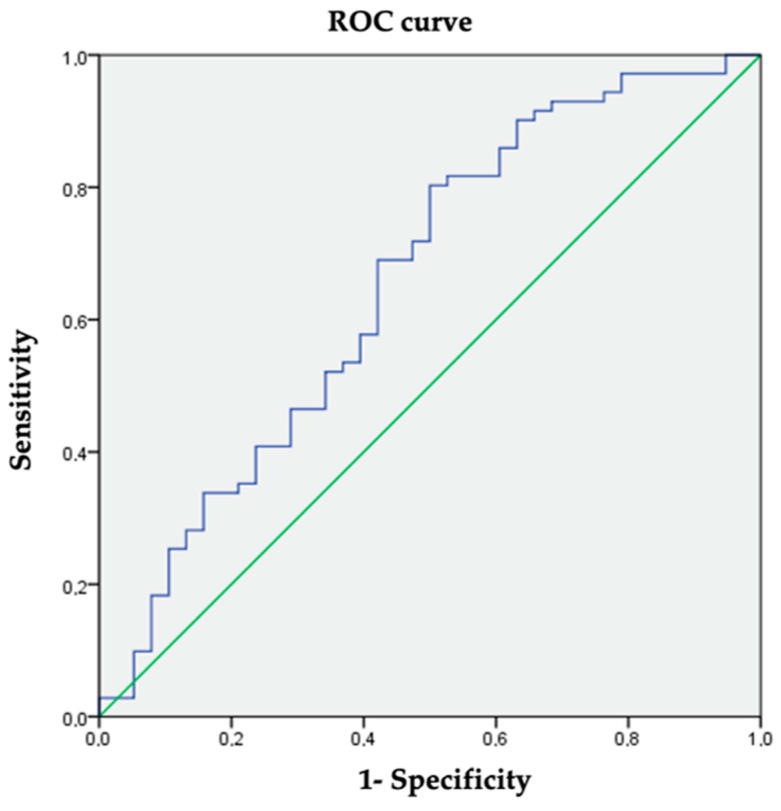
ROC curve coordinates for FIB-4 as a diagnosis of hepatic steatosis.

**Figure 4 nutrients-16-01996-f004:**
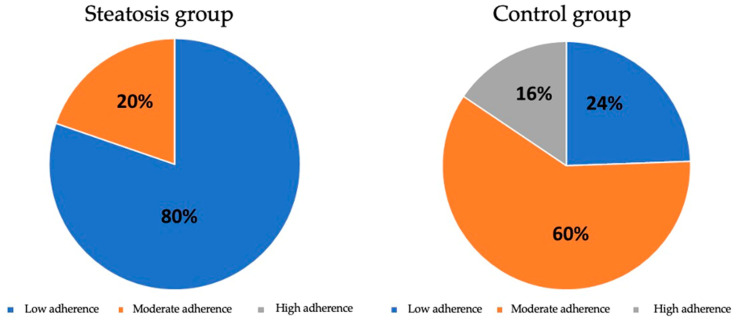
Distribution of adherence to the Mediterranean diet between groups.

**Figure 5 nutrients-16-01996-f005:**
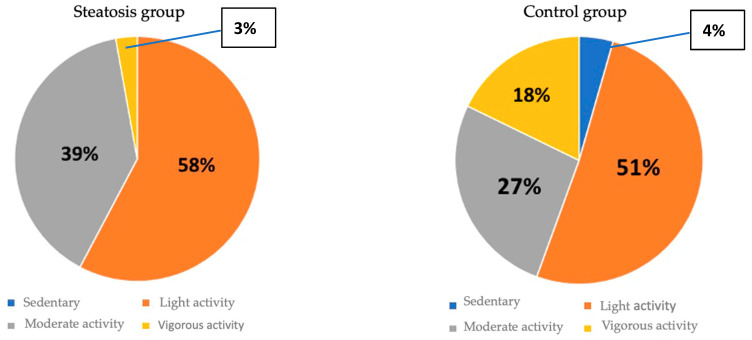
Distribution of physical activity between groups.

**Table 1 nutrients-16-01996-t001:** Distribution of anthropometric variables between groups.

	Total	CG	SG	*p*
Weight (Kg)	86.6 ± 14.0	78.5 ± 14.9	91.7 ± 10.7	0.000
IMC (Kg/m^2^)	31.4 ± 4.3	28.5 ± 4.3	33.2 ± 3.2	0.000
Adiposity (%)	39.2 ± 7.3	37.0 ± 7.4	40.6 ± 6.9	0.009

**Table 2 nutrients-16-01996-t002:** Association between obesity and hepatic steatosis according to different diagnostic criteria for obesity.

	Steatosis Group (*n* = 71)	Control Group (*n* = 45)	*p*
Global obesity according to BMI (%)	90.1%	31.1%	0.000
Global obesity according to adiposity (%)	100%	88.9%	0.008

**Table 3 nutrients-16-01996-t003:** Physical activity.

	Frequency	Percentage
Sedentary	2	1.7
Light activity	64	55.2
Moderate activity	40	34.5
Vigorous activity	10	8.6

**Table 4 nutrients-16-01996-t004:** Values of biochemical markers of steatosis in both groups.

	Ultrasound Steatosis	Mean	Standard Deviation	*p*
HSI	No	38.3262	5.50273	0.000
	Yes	43.6782	5.23534	
APRI	No	0.4568	0.21902	0.081
	Yes	0.5569	0.31134	
FIB_4	No	0.3006	0.19029	0.039
	Yes	0.3997	0.25703	

## Data Availability

Data are unavailable due to privacy or ethical restrictions.

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
