# Peer review of "Influence of Adherence to the Mediterranean Diet and Level of Physical Activity with Liver Steatosis in People Aged > 50 Years and with a BMI > 25 kg/m2: Association with Biochemical Markers"

_nutrients, 2024, doi:10.3390/nu16131996_

Round 1

Reviewer 1 Report

Comments and Suggestions for Authors

This distinguished papers examines the effect of the Mediterranean Diet on MAFLD - MASH. The statistics are strong, and the results are succinct and straightforward. I have some comments:

Results:

- Table 3: the number of subjects on a sedentary lifestyle is extremely low. Was this some sub-population?

- The sensitivity of the biomarkers does allow for screening, but their specificity is too low to replace either imaging or biopsy.

Discussion:

As the Mediterranean Diet does not fit everybody's preferences, the Atlantic Diet could be a valuable alternative.

Comments on the Quality of English Language

Some grammatical errors throughout the text.

Author Response

Comment 1: - Table 3: the number of subjects on a sedentary lifestyle is extremely low. Was this some sub-population?

RESPONSE: These patients do not belong to a subpopulation, but physical activity was estimated by means of IPAQ questionnaire and the patients probably overestimate their activity. The following paragraph has been added in a Limitations section at the end of the Discussion “Among the limitations of this study, it should be mentioned that both adherence to the Mediterranean diet and the degree of physical activity were estimated by means of questionnaires. This may lead the patient to overestimate both circumstances. In the present study it is striking that only 2 patients described physical activity within the sedentary range, which seems very unlikely in a study population over 50 years of age and with a BMI above 25 kg/m2. Therefore, extrapolation of the results obtained should be done with caution.” (4.1 Limitations).

Comment 2: The sensitivity of the biomarkers does allow for screening, but their specificity is too low to replace either imaging or biopsy.

RESPONSE: We totally agree with this affirmation and we have added it “. However, the sensitivity of the biomarkers does allow for screening, but their specificity is too low to replace either imaging or biopsy.” (Discussion last paragraph)

Comment 3: As the Mediterranean Diet does not fit everybody's preferences, the Atlantic Diet could be a valuable alternative.

RESPONSE: The following paragraph has been added “As the Mediterranean Diet does not fit everybody's preferences, the Atlantic Diet could be a valuable alternative. The fundamental difference between the Mediterranean diet and the Atlantic diet is that the percentage of carbohydrates consumed in the Mediterranean diet decreases significantly and the consumption of high quality protein increases, with the consumption of fish and seafood. These two fresh products are more interesting than meat to increase protein intake, as they also have very beneficial unsaturated fats and their caloric intake is lower. To date there have been few studies specifically looking at the Atlantic diet, so further research is needed to confirm the equivalent benefit of the Mediterranean diet.” (Discussion, 7th paragraph).

Comment 4: Some grammatical errors throughout the text.

RESPONSE: Grammar revision has been conducted by a native English speaker.

Reviewer 2 Report

Comments and Suggestions for Authors

The title accurately reflects the study's content, focusing on the relationship between diet, physical activity, and liver steatosis in an older, overweight population.

The introduction could benefit from a brief discussion on why the specific age and BMI criteria were chosen for the study population.

Results: 

  • The description of the ROC curve analysis could be expanded to explain its significance better.
  • It would be helpful to include a discussion on the potential implications of the finding that no significant differences were found in APRI values between groups.
  • The manuscript provides valuable insights into the relationship between diet, physical activity, and liver steatosis in an older, overweight population. It is well-structured and presents its findings clearly. Addressing the weaknesses mentioned above will enhance the manuscript's clarity and impact.

Author Response

Comment 1: The introduction could benefit from a brief discussion on why the specific age and BMI criteria were chosen for the study population.

RESPONSE: The following sentence has been added “As previously mentioned, excess weight is the main cause of hepatic steatosis in our environment and its prevalence increases from the 5th decade of life onwards, both in men and women. For this reason, we chose to study a population in this age range and with this excess weight [1,2].” (Introduction, last paragraph).

Comment 2: The description of the ROC curve analysis could be expanded to explain its significance better.

RESPONSE: It has been added as explanatory text for the ROC curves that the cut-off values were established according to the coordinates of the ROC curve, in order to obtain the most accurate values for sensitivity and specificity, and the most clinically meaningful. In addition, the caption of figures 2 and 3 has been changed to “ROC curve coordinates for HSI to diagnose hepatic steatosis”.

Comment 3: It would be helpful to include a discussion on the potential implications of the finding that no significant differences were found in APRI values between groups.

RESPONSE: The following sentence has been added “No significant differences in APRI values were found between the groups, but this may be attributed to the small sample size. Studies with larger numbers of individuals will be necessary to determine the potential value of the APRI biomarker as a screening for hepatic steatosis.” (Discussion, last paragraph)